# Exosome-mediated chemotaxis optimizes leader-follower cell migration

Louis González, Andrew Mugler[ID]*

Department of Physics and Astronomy, University of Pittsburgh, Pittsburgh, Pennsylvania, United States of America

* andrew.mugler@pitt.edu

## Abstract

Cells frequently employ extracellular vesicles, or exosomes, to signal across long distances and coordinate collective actions. Exosomes diffuse slowly, can be actively degraded, and contain stochastic amounts of molecular cargo. These features raise the question of the efficacy of exosomes as a directional signal, but this question has not be systematically investigated. We develop a theoretical and computational approach to quantify the limits of exosome-mediated chemotaxis at the individual cell level. In our model, a leader cell secretes exosomes, which diffuse in the extracellular space, and a follower cell guides its migration by integrating discrete exosome detections over a finite memory window. We combine analytical calculations and stochastic simulations and show that the chemotactic velocity exhibits a non-monotonic dependence on the exosome cargo size. Small exosomes produce frequent but weak signals, whereas large exosomes produce strong but infrequent encounters. In the presence of nonlinear signal transduction, this tradeoff leads to an optimal cargo size that maximizes information throughput, as quantified by the average speed of the follower cell. Using a reduced one-dimensional model, we derive closed-form expressions coupling the optimal cargo size to follower speed as a function of secretion rate, memory time, and detection sensitivity. These results identify molecular packaging and memory integration as key determinants of exosome-mediated information transmission and highlight general design principles for optimization of migration under guidance by discrete and diffusible signaling particles.

## Author summary

For cells to be capable of performing essential activities as collectives and clusters, they must be capable of communicating with each other efficiently. One way they do this is by releasing nanoparticles, known as exosomes, that transport cargos of signaling molecules. Unlike free molecules secreted directly from the

**Data availability statement:** All code is available at https://github.com/ gonzalezlouis/exosome-chemotaxis.

**Funding:** L.G. and A.M. were supported by National Science Foundation Grant No. PHY-2118561. The funders had no role in study design, data collection and analysis, decision to publish, or preparation of the manuscript.

**Competing interests:** The authors have declared that no competing interests exist.

cell, which diffuse continuously and widely, exosomes diffuse slowly, degrade over time, and contain random amounts of cargo. How do cells make reliable decisions from such noisy and intermittent information? In this study, we employed mathematical modeling and computational simulations to explore how a follower cell can faithfully track a leader cell by utilizing exosomes secreted from the latter. We found that, for a follower cell to move more reliably, exosomes ought to be of intermediate size. Too small and they transmit little information. Too large and they are released too infrequently. This balance suggests a general principle by which cells can bundle information in order to communicate effectively.

## Introduction

Cells often rely on long-range communication to coordinate migration, differentiation, and collective behavior. Intercellular signaling involves extracellular vesicles, also known as exosomes, which are nanoscale lipid-bound particles (30 to 150 nm) secreted into the extracellular environment [1,2]. These exosomes carry a variety of molecular cargoes, including proteins, lipids, mRNAs, and microRNAs, and can influence gene expression and cell behavior after uptake by recipient cells [3–5]. Although exosomes have been widely studied in the context of cancer, immune modulation, and tissue development [6–9], recent work suggests that they may also play a role in guiding cell migration. For instance, amoeba cells have been shown to secrete exosomes containing the chemoattractant cAMP that guide the migration of other cells [10]. Similarly, cancer cells have been shown to migrate in response to exosomes, and to do so less efficiently when the exosomes are depleted of their chemical content [11,12].

Chemotaxis, the process by which cells move in response to external signals, is a fundamental mechanism in development, immunity, and disease [13]. Traditional models of chemotaxis involve small diffusible molecules that bind receptors on the cell surface, allowing the cell to detect gradients and bias its movement accordingly [14,15]. However, signaling through exosomes introduces several new features that distinguish it from conventional chemotactic models. Exosomes diffuse more slowly than small molecules, usually carry multiple signaling molecules in a single packet, and are subject to degradation or clearance over time [16–18]. In addition, the number of molecules within each exosome is inherently stochastic due to the random nature of exosome formation and cargo loading [16,17]. Experimental quantification reveals that cargo load can vary significantly: while some exosomes may carry sparse payloads (e.g., <1 miRNA copy per vesicle on average [19]), others are enriched with high concentrations of signaling proteins such as chemokines [20]. This wide variability necessitates a theoretical approach that treats the mean cargo size as a tunable parameter. Signaling molecules are found both within and on the surface of exosomes [20,21]; in either case, signaling molecules arrive at the receiving cell in discrete packets.

These features are particularly relevant in the context of tumors, where exosome-based signaling plays a central role in modulating the tumor microenvironment,

establishing premetastatic niches, and coordinating collective invasion [22–24]. For example, tumor-derived exosomes have been shown to enhance the migratory behavior of both cancer cells and surrounding stromal cells by delivering specific integrins, growth factors, or chemokines that remodel local tissue and establish chemotactic gradients [25,26]. Some studies report that metastatic cells preferentially secrete exosomes enriched in molecules that promote matrix remodeling and immune evasion [23,24]. In this way, exosomes not only carry positional information, but also participate in shaping the spatial structure of the signaling landscape. This coupling between signal production, transport, and environmental feedback is fundamentally different from passive gradient sensing and may be the basis for the complex migration patterns observed in metastatic progression. These observations motivate the need to quantitatively characterize how exosome-mediated signaling constrains or enhances the ability of a cell to detect gradients and coordinate movement within heterogeneous tissues.

To date, models of exosome-mediated communication have mainly focused on population-scale effects or systems-level regulation [18,22]. However, relatively few studies have addressed the statistical and physical constraints that govern exosome-based chemotaxis at the single-cell level. How do the discrete and stochastic nature of exosome secretion, transport, and degradation shape the sensory landscape of a recipient cell? What are the trade-offs between exosome cargo size, secretion rate, and detection fidelity? To what extent can cells integrate exosome encounters to produce a net directional bias in migration?

In this study, we develop a theoretical and computational framework to explore how a migrating cell can use exosome detections to guide its motion. We consider a model in which a leader cell secretes exosomes that diffuse through the extracellular space and degrade over time. A follower cell updates its otherwise random direction based on exosome detection events. By combining stochastic models of exosome secretion, molecular packaging, diffusion, and degradation with a memory-based detection mechanism, we derive analytical predictions for the follower cell's chemotactic velocity. These predictions are compared with stochastic simulations to identify the key factors that govern chemotactic efficiency. In particular, we find an optimal exosome size and diffusion coefficient that maximizes the follower cell's velocity. Our results reveal how the interplay between molecular packaging, detection thresholds, and memory duration sets the precision and speed of exosome-guided migration. In doing so, we provide a quantitative framework for understanding exosome-based communication from the perspective of information processing in single cells.

## Methods

### Model description

To understand how migrating cells can extract directional information from diffusible exosomes, we construct a minimal 2D model in which a "follower" cell tracks a "leader" cell that secretes exosomes (Fig 1a). These exosomes serve as discrete stochastic carriers of molecular signals. Rather than responding to a continuous chemical gradient, the follower detects and integrates molecular encounters over time to determine whether to move in a directed or undirected fashion. The goal of this model is to capture the biophysical constraints associated with exosome-based signaling (specifically, the noise and sparsity introduced by finite exosome numbers, stochastic packaging, slow diffusion, and finite memory integration) and to determine how they impact the reliability of directional migration.

We model the leader cell as a point source moving at a constant velocity $v_0$. It secretes signaling molecules at a fixed rate $\nu$ (molecules per unit time), representing its available communication budget. Rather than releasing molecules individually, the leader packages them into discrete exosomes each carrying a random molecular cargo. The number of molecules $n_i$ within each exosome is drawn from a Poisson distribution with mean $\bar{n}$:

$$n_i \sim \text{Poisson}(\bar{n}). \tag{1}$$

PLOS Computational Biology

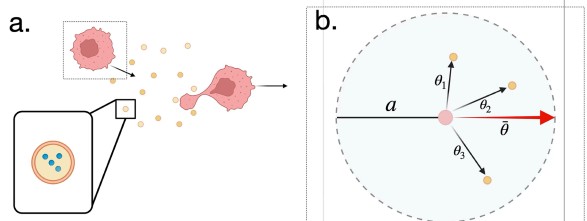

**Fig 1**. **Biological sketch of exosome-amplified chemotaxis [27].** (a) The chemoattractant (blue) produces a spatial gradient that the leading cell senses and migrates towards. In its wake, the leader cell secretes the same chemoattractant packaged in exosomes (yellow). The exosomes diffuse at a slower rate than the individual molecules and produces a secondary spatial gradient of the same chemoattractant. The follower cell then migrates towards the leader, and indirectly towards the primary gradient, by sensing the exosomes. (b) Direction sensing mechanism: The follower cell has a capture radius $a$. Exosomes detected within this radius are recorded as arrival vectors (thin arrows) defined by the angle $\theta_i$ relative to the cell center. The cell integrates these inputs to compute a weighted average direction $\bar{\theta}$ (thick red arrow) for migration.

This introduces noise in the form of variable packet sizes. Although the total molecular flux remains fixed at $\nu$, the exosome secretion rate becomes $\nu/\bar{n}$, reflecting a tradeoff between frequency and payload size.

Once secreted, exosomes undergo two-dimensional diffusion with diffusivity $D$. The position $\big(x_i(t), y_i(t)\big)$ of the $i$-th exosome evolves via independent Brownian displacements in each Cartesian direction:

$$x_i(t + \Delta t) = x_i(t) + \sqrt{2D\,\Delta t}\,\eta_{x,i}(t), \tag{2}$$

$$y_i(t + \Delta t) = y_i(t) + \sqrt{2D\,\Delta t}\,\eta_{y,i}(t), \tag{3}$$

where $\eta_{x,i}$ and $\eta_{y,i}$ are independent standard normal variates at each time step $\Delta t$. This formulation captures passive, isotropic diffusion of exosomes in the extracellular milieu; extensions to include advective flow or matrix anisotropy can be incorporated as needed.

We model exosome degradation (for example, by enzymatic breakdown or clearance) as a first-order decay process with timescale $\mathcal{T}$. That is, in each time step $\Delta t$, each exosome has probability $\Delta t/\mathcal{T}$ of degrading.

The follower cell detects exosomes that reach within a capture radius $a$ of the its position $(x_f, y_f)$ (Fig 1b). This effectively treats the follower cell as having an absorbing boundary of size $a$. Whether the exosome binds to surface receptors or fuses, the interaction occurs at a specific location on the follower cell's membrane. Therefore, we assume that the cell has a mechanism to retain and integrate information about the arrival location of the exosome, similar to how cells retain and integrate that information for single-molecule chemoreception.

Each detection event is characterized by the exosome's cargo size $n_i$, its arrival time $t_i$, and the angle of its arrival $\theta_i$. The cargo size is processed through a nonlinear activation function

$$\alpha_i = \frac{n_i^H}{n_i^H + K^H}, \tag{4}$$

where $K$ is a detection threshold and $H$ is the Hill coefficient that modulates the sharpness of the response. This formulation captures the ultrasensitive biochemical activation characteristic of cellular signaling cascades.

The arrival time is processed through the cell's memory, which decays exponentially with a timescale $\tau$,

$$m_i(t) = e^{-(t-t_i)/\tau}. \tag{5}$$

This prescription limits the effective duration of the signaling induced by each exosome without invoking explicit decay kinetics. The integrated memory of all events is then

$$M(t) = \sum_i \alpha_i m_i(t), \tag{6}$$

where the sum is taken over all previously detected exosomes. This leaky integration models the transient nature of intra-cellular signaling and allows for time-dependent accumulation and eventual loss of information.

The arrival angle is defined geometrically at the moment of detection as $\theta_i = \arctan((y_i - y_f)/(x_i - x_f))$, where $(x_i, y_i)$ is the exosome position and $(x_f, y_f)$ is the follower cell position (Fig 1b). This angle represents the direction on the cell membrane where the exosome signal is received. The arrival angles are processed by a weighted average, with the weights given by the activation strength and memory of each event:

$$\bar{\theta}(t) = \tan^{-1}\left[\frac{\sum_i \alpha_i m_i(t) \sin \theta_i}{\sum_i \alpha_i m_i(t) \cos \theta_i}\right]. \tag{7}$$

At each timestep $\Delta t$, the direction of motion is then sampled according to

$$\theta(t) = \begin{cases} \bar{\theta}(t) & \text{with probability } M(t)/[1 + M(t)] \\ \text{Uniform}[0, 2\pi) & \text{otherwise.} \end{cases} \tag{8}$$

This form ensures motion is biased toward recent sources of high signal. Specifically, when $M(t) \gg 1$, the follower is biased toward the angles of recent detection events via $\bar{\theta}(t)$. Conversely, when $M(t) \ll 1$, the follower performs an unbiased random walk. This can occur either if no signal has been detected in some time ($m \ll 1$), or if signals are weak ($\alpha \ll 1$). Altogether, this mechanism functionally mimics the response to an internal polarity gradient informed by the history of localized detection events [28].

The follower position is then updated along the chosen direction as

$$x_f(t + \Delta t) = x_f(t) + v_0 \Delta t \cos \theta(t), \tag{9}$$

$$y_f(t + \Delta t) = y_f(t) + v_0 \Delta t \sin \theta(t), \tag{10}$$

where the follower speed $v_0$ is the same as the leader speed.

## Simulation procedure and parameter values

Simulations are carried out in discrete time with a fixed step size $\Delta t$. The leader is initialized 10 $\mu$m ahead of the follower, which is placed at the origin. The dynamics, detection, and degradation of exosomes are simulated according to the rules defined above.

Each simulation is carried out for a total time $T$, and the follower's position is recorded at each time step. For each set of parameters, we perform multiple independent simulations (typically $10^3$–$10^4$ runs) and compute the ensemble average of the follower's displacement under a total simulation time $T$:

$$\bar{v} = \frac{\langle x_f(T) - x_f(0) \rangle}{T}. \tag{11}$$

The resulting migration velocity characterizes the efficacy of chemotaxis under different secretion, detection, and memory regimes.

To resolve the relevant physical dynamics while maintaining computational efficiency, we simulate exosome diffusion and follower cell motion using separate time steps: $\Delta t_{\text{exo}}$ for exosome dynamics and $\Delta t_{\text{cell}}$ for cell updates. Exosomes have diffusivities on the order of $D \sim 10^2 - 10^3$ $\mu\text{m}^2/\text{min}$ [29], while typical cell migration speeds range from $v_0 \sim 0.1$ to 1 $\mu\text{m/min}$ [30]. Because exosomes diffuse much more rapidly than cells migrate ($a^2/D \ll a/v_0$), we choose $\Delta t_{\text{exo}} \ll \Delta t_{\text{cell}}$ to accurately resolve the stochastic fluctuations and spatial distribution of exosomes.

Memory timescales of eukaryotic cells during migration can range from minutes to days, although short-term memory as measured by directional persistence is typically tens of minutes [31,32]. Therefore we take $\tau = 5$ minutes and we later extend this up to approximately an hour (see Fig 3d later on).

The parameters used throughout the work, as well as the rationale for their values, are listed in Table 1. The code is made publicly available at [33].

## Results

### Average migration velocity depends non-monotonically on exosome cargo size

We first asked how the molecular granularity of exosome packaging affects the efficiency of follower cell migration. In our model, a fixed molecular secretion rate $\nu$ is distributed across exosomes with Poisson-distributed cargo sizes of mean $\bar{n}$. This implies that increasing $\bar{n}$ reduces the number of exosomes per unit time while increasing the molecular content per exosome. Since the follower cell makes decisions based on temporally integrated molecular detections, this tradeoff influences both the frequency and reliability of directional updates.

Fig 2 shows the average migration velocity $\bar{v}(\bar{n})$ of the follower cell as a function of the mean exosome cargo size. We see that for Hill coefficient $H = 1$ (a), the follower cell is fastest for single-molecule diffusion ($\bar{n} = 1$), whereas for larger $H$ (b), there is an optimal size $\bar{n}$ that maximizes the follower velocity. We find that the location of the optimum is generally set by $\bar{n} \approx K$, and that apart from this feature, changing $K$ does not affect the results qualitatively; therefore we set $K = 25$ molecules throughout. The optimum at larger $H$ arises from the switch-like nature of the activation function (Eq 4): for $\bar{n} < K$, the cargo is insufficient to trigger the activation; whereas for $\bar{n} > K$, the response saturates, and further increasing the cargo size implies a penalty in exosome frequency. Thus, the optimum represents the most efficient packaging size that reliably exceeds the detection threshold. We also see in Fig 2 that reducing the decay time decreases the follower speed but otherwise has no qualitative effect; therefore we take $\mathcal{T} \to \infty$ from here on.

### Approximate 1D model explains the simulation behavior

To more quantitatively understand the optimum in Fig 2, and to understand the role of model parameters, we develop an effective 1D model that is analytically solvable (Fig 3(a), top). In this model, the leader cell moves at constant speed $v_0$,

**Table 1**. Parameters used in this work, and their values except when varied.

| | Meaning | Value | Rationale |
|---|---|---|---|
| $a$ | Cell length scale | 10 $\mu\text{m}$ | [34–37] |
| $\nu$ | Exosome secretion rate | 350 molecules/min | [38] |
| $v_0$ | Cell velocity | 3.4 $\mu\text{m/hr}$ | [39] |
| $D$ | Typical exosome diffusivity | 250 $\mu\text{m}^2/\text{min}$ | [29] |
| $\tau$ | Memory time | 5 minutes | [31,32] |
| $K$ | Molecule threshold | 25 molecules | No qualitative change |
| $\mathcal{T}$ | Exosome degradation time | $\infty$ | No qualitative change |
| $\Delta t_{\text{exo}}$ | Time step, exosomes | 0.1 minutes | $\Delta t_{\text{exo}} \ll a^2/D$ |
| $\Delta t_{\text{cell}}$ | Time step, cell | 40 minutes | $\Delta t_{\text{cell}} \ll a/v_0$ |

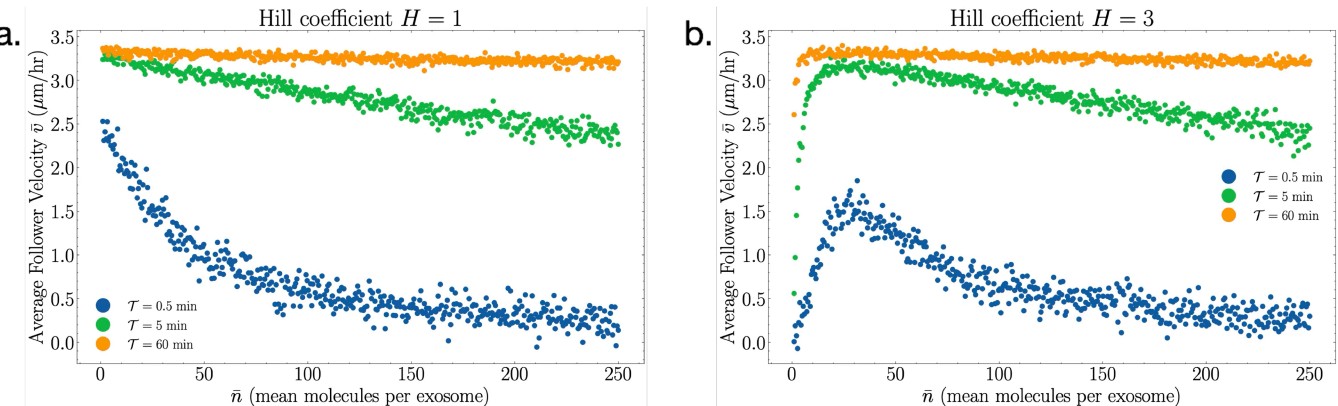

**Fig 2**. **Average velocity of the following cell as a function of the mean molecule cargo amount per secreted exosome for three sample exosome degradation times $\mathcal{T} = 0.5, 5$, and 60 minutes.** The two response regimes are plotted: (a) $H = 1$ and (b) $H>1$ (shown as $H = 3$). Parameters used are enumerated in Table 1.

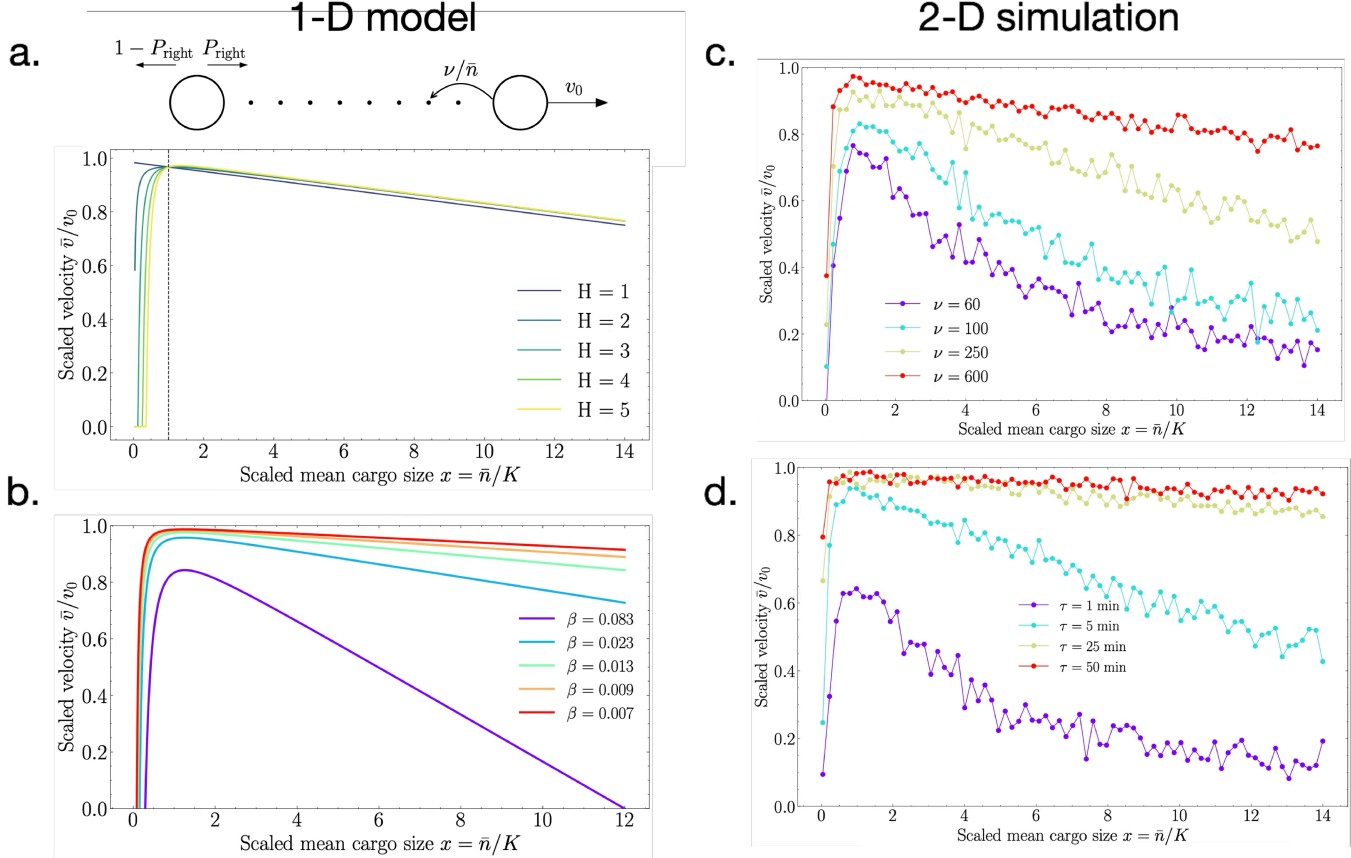

**Fig 3**. **Comparison of the approximate 1D model (a, b) with the 2D simulation (c, d).** In the 1D model, the leader cell moves at constant speed $v_0$ and leaves behind a trail of non-diffusing ($D = 0$) exosomes. The follower cell moves to the left or right, with the probability $P_{\text{right}}$ increasing with detection events. The average follower velocity (a) exhibits a maximum with cargo size for $H>1$ as in the 2D simulations (Fig 2) and (b) decreases with the composite parameter $\beta = K/\nu\tau$. The 2D simulations confirm the dependence on $\beta$ through (c) secretion rate $\nu$ and (d) memory time $\tau$.

depositing exosomes along its path. For simplicity, we assume here that the exosomes have zero diffusivity ($D = 0$). This forms a trail of exosomes, each carrying a Poisson distributed number of molecules with mean $\bar{n}$, and the total molecular output is fixed at a constant rate $\nu$. As the follower cell performs a random walk along this trail, it encounters these exosomes and evaluates whether to bias its motion toward the leader on the basis of accumulated chemical information.

The one-dimensional analog of Eq 8 is

$$P_{\text{right}} = \frac{1/2 + M(t)}{1 + M(t)}, \tag{12}$$

where $P_{\text{right}}$ is the probability that the follower cell moves to the right (toward the leader cell), and $M(t)$ is as defined in Eq 6. We see in Eq 12 that, as before, when $M(t) \gg 1$ the follower is biased toward recent detection events ($P_{\text{right}} \to 1$), and when $M(t) \ll 1$ the follower performs an unbiased random walk ($P_{\text{right}} \to 1/2$).

To simplify $M(t)$ in this model, we approximate each exosome as providing the activation strength of the average cargo size,

$$M(t) = \sum_i \alpha_i m_i(t) \approx \bar{\alpha} N(t), \tag{13}$$

where

$$\bar{\alpha} = \frac{\bar{n}^H}{\bar{n}^H + K^H}, \tag{14}$$

and $N(t) = \sum_i m(t)$ is the number of exosomes that the follower has encountered within its latest memory window $\tau$. This number can be approximated as a ratio of lengths: the length the follower travels in time $\tau$ at its (as yet unknown) average velocity $\bar{v}$, and the length between exosomes dropped at rate $\nu/\bar{n}$ by the leader moving at speed $v_0$. Thus,

$$N(t) \approx \frac{\bar{v}\tau}{v_0/(\nu/\bar{n})} = \frac{\bar{v}\tau\nu}{v_0\bar{n}}. \tag{15}$$

The average follower velocity, in turn, is determined by $P_{\text{right}}$ as

$$\bar{v} = (2P_{\text{right}} - 1)v_0, \tag{16}$$

which describes the drift velocity of a biased random walk with individual step speed $v_0$. Because $P_{\text{right}}$ (Eq 12) depends on $\bar{v}$ via Eqs 13–15, we solve Eq 16 for $\bar{v}$ self-consistently. The result is

$$\frac{\bar{v}}{v_0} = \begin{cases} 1 - \phi & \text{if } \phi < 1, \\ 0 & \text{otherwise,} \end{cases} \tag{17}$$

where

$$\phi = \beta x(1 + x^{-H}), \tag{18}$$

and we have defined

$$x \equiv \frac{\bar{n}}{K} \quad \text{and} \quad \beta \equiv \frac{K}{\tau\nu}. \tag{19}$$

This result shows that the follower moves only if the exosome frequency and signal strength are large enough to keep $\phi < 1$.

Eq 17 is plotted in Fig 3a and we see that the average follower velocity exhibits the same behaviors with $H$ that we saw in the 2D simulations (Fig 2): for $H = 1$ the velocity decreases monotonically with cargo size $\bar{n}$, whereas for $H > 1$ the velocity has a maximum as a function of cargo size $\bar{n}$. Differentiating Eq 17, we find that the maximum occurs at

$$\frac{\bar{n}^*}{K} = (H-1)^{1/H} \xrightarrow[H \gg 1]{} 1. \tag{20}$$

This result explains why the simulations generally showed that the optimum occurs at $\bar{n} \approx K$. The maximum makes sense because smaller $\bar{n}$ values do not strongly trigger activation (Eq 14), while large $\bar{n}$ values limit the number of exosome encounters (Eq 15).

At the optimum, the maximum velocity is

$$\frac{\bar{v}_{max}}{v_0} = 1 - \beta \frac{H}{H-1}(H-1)^{1/H} \xrightarrow[H \gg 1]{} 1 - \beta. \tag{21}$$

Eq 17 is plotted for different $\beta$ values in Fig 3b, and indeed we see that the maximum decreases with $\beta$. Given the definition of $\beta = K/(\tau\nu)$ (Eq 19), this finding implies that the maximum follower velocity should decrease with the detection threshold $K$, increase with the memory time $\tau$, and increase with the secretion rate $\nu$. These dependences make sense: the follower velocity (i) decreases with $K$ because then detection events are less likely to trigger activation; (ii) it increases with $\tau$ because then the follower cell remembers the detection events for longer; and (iii) it increases with $\nu$ because then the leader cell secretes more signaling molecules. Thus, our approximate 1D model elucidates the physical determinants of efficient chemotaxis. It establishes $\beta$ as a unifying parameter that quantifies the trade-off between signal availability and temporal integration in exosome-guided chemotaxis.

Fig 3c and 3d confirm that these predictions hold in our 2D simulations. Specifically, Fig 3c demonstrates that the average follower velocity increases with the secretion rate $\nu$, and Fig 3c demonstrates that the average follower velocity increases with the memory time $\tau$. We see that the changes in Fig 3c and 3d are qualitatively similar to the changes in Fig 3b.

## Average migration velocity depends non-monotonically on exosome diffusion

We next asked how the chemotactic efficiency depends on the diffusivity of the exosomes. We suspected that there might be a tradeoff because both extremes are suboptimal. On the one hand, if the exosomes do not diffuse at all, the follower cell may never encounter them, especially in higher spatial dimensions. On the other hand, if they diffuse extremely rapidly, their directional information would be lost immediately.

To quantify this trade-off, we sampled a range of diffusivities $D$ from 0 to 1000 $\mu m^2$/min. Fig 4a presents representative trajectories of the follower cell at different $D$ values after $T = 1440$ min (one day; placed at the origin for illustration). Immobile exosomes ($D = 0$) yield very little net displacement: Without spatial diffusion, follower encounters are rare and highly stochastic. At very high diffusivity ($D > 500$ $\mu m^2$/min), trajectories revert to random walks, as rapid dispersion eliminates the directional signal. At intermediate values ($D \approx 50 - 150$ $\mu m^2$/min), trajectories are most directed, indicating the greatest chemotactic response. These simulations also reproduced the most physiologically realistic displacement: for $T = 1440$ min and leader velocity $v_0 = 3.4$ $\mu m$/hr, follower cells occasionally reached $\sim 3000$ $\mu m$ (3 mm), matching observed migration distances in vitro and in vivo (see review by Sung et al. [12]).

We further assessed the full $D$-dependence of the chemotactic velocity in Fig 4b by varying the diffusivity from 0 to 1000 $\mu m^2$/min, simulating 2D random walks over various total durations $T$. The peak velocity consistently occurred near $D \approx 100$ $\mu m^2$/min, independent of $T$, although longer simulations (e.g., $T = 1440$ min or 24 h) reduced the trajectory-to-trajectory variability. When $T$ is extended to one day, the optimal diffusivity remained unchanged, supporting the robustness of this finding.

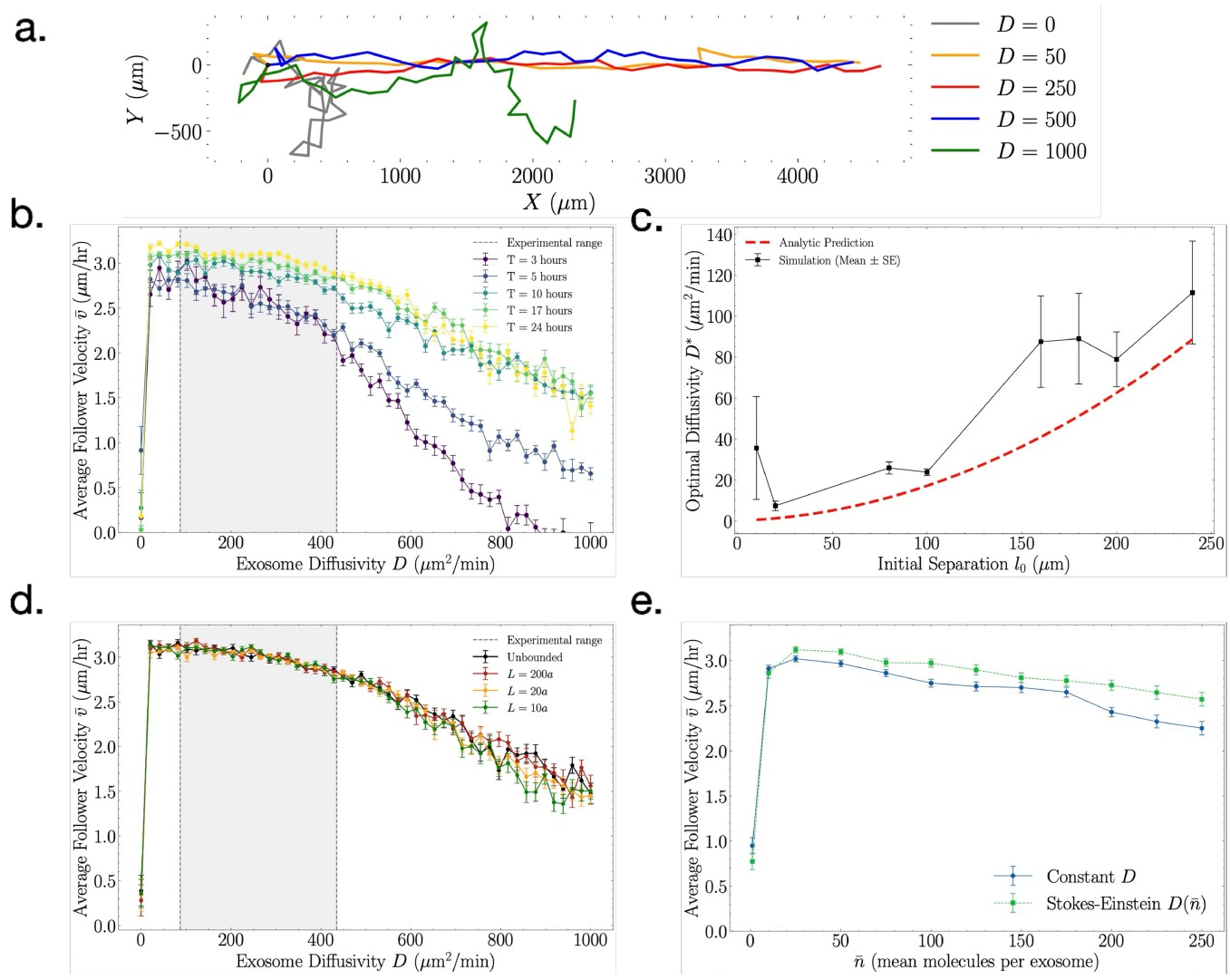

**Fig 4**. **Effects of exosome diffusivity on follower cell chemotaxis.** (a) Simulation trajectories of the follower cell with various values of exosome diffusivity $D$ after $T = 1$ day (1440 minutes). (b) Velocity as a function of diffusivity for various total simulation times $T$. Overlaid is the physiological range of expected diffusivity using experimentally measured exosome size $a = 30 - 150$ nm. (c) Optimal diffusivity as a function of initial cell separation (black; mean and standard error over 5 trials), compared with prediction in Eq 23 (red). (d) As in b, for $T = 24$ h, but with two reflective boundaries (to both follower cell and exosomes) placed parallel to the leader cell's movement direction, at various distances $L$, shown in multiples of the cell length-scale $a$. (e) Velocity as a function of mean cargo load $\bar{n}$ for constant diffusivity $D$ (blue) and a diffusivity that scales with cargo load calculated using the Stokes-Einstein equation, $D(\bar{n}) = D_0(\bar{n}_0/\bar{n})^{1/3}$ (green). Here $D_0 = 300$ $\mu$m$^2$/min and $\bar{n}_0 = 50$ molecules.

Physiologically relevant diffusivities for exosomes of size $a_{\text{exo}} = 30 - 150$ nm can be estimated using the Stokes–Einstein relation $D = k_B T/6\pi\eta a_{\text{exo}}$, which yields $D \sim 90 - 440$ $\mu$m$^2$/min in aqueous environments at room temperature (gray region in Fig 4b). This range aligns closely with the optimal diffusivity observed in our simulations, implying that typical exosome sizes support efficient chemotactic signaling [10,11,40,41]. In fact, we see in Fig 4b that the follower cell velocity remains high even below this range, suggesting that chemotaxis would remain efficient even if exosome diffusion is reduced by interactions with the the extracellular matrix.

These results demonstrate that diffusivity serves as a critical modulator, dictating whether exosome transport enhances or impairs signal fidelity. Importantly, diffusivity interacts with the secretion rate and the memory window to determine the range of effective chemotactic guidance. This supports the concept that exosome diffusivity must be balanced: sufficiently to reach the follower, but not so much as to obfuscate spatial information [42,43].

To understand this tradeoff more quantitatively, we develop an analytic argument as follows. The exosome signal must arrive at the follower cell by diffusion, even as the signal's source (the leader cell) moves away at speed $v_0$. Assuming that the two cells start with a separation $l_0$, this arrival time $t$ satisfies

$$\sqrt{4Dt} = l_0 + v_0 t, \tag{22}$$

where $\sqrt{4Dt}$ is the typical distance covered by the diffusing exosomes, and $l_0 + v_0 t$ is the distance covered by the leader cell. We hypothesize that optimal diffusion sets this arrival time to maximize the signal difference across the follower cell's lengthscale $a$. There are two ways to obtain a timescale from $a$ using the parameters above: $t \sim a/v_0$ or $t \sim a^2/D$. Inserting either $t = a/v_0$ or $t = a^2/D$ into Eq 22 and solving for the diffusivity obtains, respectively,

$$D^* = \frac{v_0}{4a}(l_0 + a)^2 \tag{23}$$

or $D^* = v_0 a^2/(2a - l_0)$. The latter is negative since $l_0 > 2a$, and therefore only the former, Eq 23, makes physical sense. Whereas $a$ and $v_0$ are biophysical parameters that are experimentally well characterized (Table 1), the initial cell separation $l_0$ can vary. Eq 23 predicts that the optimal exosome diffusivity increases with initial cell separation, which makes sense because otherwise the signal would not reach the follower cell before the leader cell got too far away.

To test this prediction, we ran many simulations like those in Fig 4b for various initial cell separations $l_0$. At each value of $l_0$, we determined the diffusion coefficient that maximizes the follower velocity. The results are shown by the black data points in Fig 4c. Eq 23 is plotted in red, and we see good agreement, especially since this prediction is a heuristic argument that only aims to capture the basic physics.

As an additional test of the robustness of our finding of an optimal diffuivity, we asked whether this result is affected by confinement, which is relevant in physiological environments. We added two reflective walls (for both follower cell and exosomes), placed parallel to the leader cell's movement direction, at variable distances $L$ on either side of the leader cell. As seen in Fig 4c, this modification results in little change in the follower dynamics, even as the wall distance is reduced to ten cell radii $a$. In particular, the optimal exosome diffusivity persists.

Lastly, we recognized that the physical size of an exosome may correlate with its cargo size $\bar{n}$. In this case, larger cargo sizes would correspond to slower diffusion via the Stokes-Einstein relation. Specifically, if volume scaled with cargo size, we would expect $D \sim \bar{n}^{-1/3}$. Since slower diffusion leads to faster follower velocities over most of its range (Fig 4b), we expect this effect to enhance follower velocity, particularly for larger cargo sizes. Fig 4e shows the result of simulations that incorporate this cargo-dependent diffusivity (green), compared to our original protocol that does not (blue). We see that follower velocity is indeed enhanced to a modest degree. Importantly, our previous finding, an optimal cargo size, remains robust to this modification.

## Discussion

In this work, we have developed and analyzed a minimal theoretical framework to understand how exosome–mediated signaling can guide single cell chemotaxis. By combining an analytically tractable 1D model with 2D stochastic simulations, we identified a non-monotonic dependence of the follower cell's migration velocity $\bar{v}$ on the average exosome cargo size $\bar{n}$. In the non-cooperative regime ($H = 1$), larger cargo simply weakens chemotactic bias, whereas in the cooperative

regime ($H > 1$) we found an optimal cargo size that balances the trade-off between packet frequency and signal amplitude (Figs 2 and 3).

A central finding from our analytical model is the dimensionless parameter $\beta$, which captures the interaction between the molecular secretion rate $\nu$, the detection threshold $K$, and the memory timescale $\tau$. Both the 1D model and its 2D validation demonstrate that decreasing $\beta$, either by increasing secretion ($\nu$) or by extending memory ($\tau$), monotonically enhances the chemotactic velocity until saturation (Fig 3c and 3d).

By considering the role of exosome diffusion we discover a second trade-off: if diffusion is too slow, detection is infrequent; if diffusion is too fast, detection is non-directional. This tradeoff leads to an optimal diffusion coefficient that agrees with that estimated for typical exosome sizes and corresponds to follower cell velocities that match experimental observations (Fig 4). Together with the optimal cargo size, this observation argues for exosome packaging as an efficient form of chemical communication in leader-follower contexts.

The findings in this work have two broad implications. First, the concept of optimal diffusivity suggests that exosome size and microenvironment viscosity jointly constrain the signaling range in tissues. Second, by defining exosome chemotaxis in terms of a key dimensionless parameter ($\beta$), we provide a unifying metric to compare various experimental systems and to guide the engineering of synthetic exosome-based communication in tissue engineering or drug delivery contexts.

Further studies of exosome-mediated chemotaxis in heterogeneous environments may be warranted. For example, recent work demonstrates that neutrophils can secrete DNA networks that physically ensnare LTB4-containing exosomes, thereby creating a stable, high-fidelity gradient for follower cells [44]. This mechanism effectively immobilizes the signal packets ($D \to 0$) after secretion, preventing diffusive dissipation. Extending our model to account for such spatially variable diffusivity or trapping mechanisms could reveal further design principles by which cells structure their communication landscape. In addition, coupling exosome secretion dynamics to cell-intrinsic feedback, such as up-regulation of secretion under mechanical stress, could reveal additional design principles for collective migration in development and cancer. Finally, rigorous experimental quantification of molecular fluxes and exosome payload distributions will be essential to fully constrain and validate theoretical models of exosome-mediated chemotaxis.

## Acknowledgments

We thank Bumsoo Han for helpful discussions.

## Author contributions

**Conceptualization:** Louis González, Andrew Mugler.

**Formal analysis:** Louis González, Andrew Mugler.

**Funding acquisition:** Andrew Mugler.

**Investigation:** Louis González, Andrew Mugler.

**Methodology:** Louis González, Andrew Mugler.

**Project administration:** Andrew Mugler.

**Software:** Louis González.

**Supervision:** Andrew Mugler.

**Validation:** Louis González.

**Visualization:** Louis González.

**Writing – original draft:** Louis González.

**Writing – review & editing:** Andrew Mugler.

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
