## [Decision Letter · Decision Letter 0]

11 Dec 2025

PCOMPBIOL-D-25-02085

Exosome-mediated chemotaxis optimizes leader-follower cell migration

PLOS Computational Biology

Dear Dr. Mugler,

Thank you for submitting your manuscript to PLOS Computational Biology. As you will see with the reviewer comments - the reviewers do see the novelty of this work, and appreciate the approach taken, but point out several important limitations and caveats in this study. After careful consideration, we feel that it has merit but does not fully meet PLOS Computational Biology's publication criteria as it currently stands. Therefore, we invite you to submit a revised version of the manuscript that addresses the points raised during the review process.

We look forward to receiving your revised manuscript.

Kind regards,

Sunil Laxman, PhD

Academic Editor

PLOS Computational Biology

Marc Birtwistle

Section Editor

PLOS Computational Biology

**Journal Requirements:**

At this stage, the following Authors/Authors require contributions: Louis González, and Andrew Mugler. Please ensure that the full contributions of each author are acknowledged in the "Add/Edit/Remove Authors" section of our submission form.

4) Please amend your detailed Financial Disclosure statement. This is published with the article. It must therefore be completed in full sentences and contain the exact wording you wish to be published.

**Reviewers' comments:**

Reviewer's Responses to Questions

**Comments to the Authors:**

Reviewer #1: Attached file

Reviewer #2: In this manuscript, Gonzalez and Mugler analyse the plausiblity of, and physical trade-offs inherent to, exosome-mediated directional signalling in cellular populations.

The authors have two primary results, both of which are well-supported by numerical and analytical arguments. The first is that cells face a trade-off between the number and "quality" of exosomes -- the idea being that the cell has a finite number of signalling molecules to package into exosomes, and can either send out a few exosomes that are well-packed with signalling molecules, or many exosomes with fewer signalling molecules each. If the receiving cell requires a minimum number of signalling molecules per exosome to faithfully read the incoming information, then a clear optimum exists: the sending cell should send out as many exosomes as possible while keeping the mean number of signalling molecules/exosome at the minimum required to elicit a response in the receiving cell.

The second result is more subtle, and relates the diffusivity of the exosome to its capacity to carry meaningful directional information. The trade-off is well expressed by the authors: if an exosome diffuses too quickly, then it is of little value in indicating to the receiving cell which direction it came from. On the other hand, if it diffuses too slowly, it may be unable to find and collide with the receiver cell in the first place. The authors show, by exhaustive simulation, that this is indeed the case -- an intermediate exosome diffusivity maximises the speed of the receiving cell. Notably, the experimental range of exosome diffusivities coincides with the optimal range found in simulation.

These results are certainly interesting and new. Of the two, the latter is the most striking, though it is relatively underdeveloped compared to the first result. The first result is interesting, though arguably the trade-off is (a) straightforward, and (b) the result of modelling choices that are (as written) untethered to biological details.

My major questions are:

1. Could the authors provide some molecular detail to flesh out the plausibility of the first result? Are signalling molecules (e.g., chemoattractants) indeed packed within exosomes (as opposed to packed on their surface), and what are the typical numbers (if known)? Why couldn't the receiving cell simply use exosome fusion as a directional signal, rather than relying on a noisy read-out of chemoattractant number?

2. The second part is compelling, but could benefit from some of the attention paid to the first part. For instance, could the authors determine what sets the optimal range of diffusivities? While it may not make sense to study cell migration in five spatial dimensions, how do these results change with the geometry of the space that cells move in? (e.g., in a channel of variable width -- I presume that for small enough widths, there is not downside to rapid diffusion?)

3. Finally, in the Discussion, the authors state:

"First, they demonstrate that exosome-packaged signals can rival the information content of smooth molecular gradients, provided that cargo size, secretion rate, and memory integration are properly tuned."

This is quite a strong claim, and as far as I can tell is not justified by the results in this manuscript. Could the authors provide some more context and/or comparision of the two modes of chemotaxis?

In summary, this is an interesting modelling paper that, with some changes, would be a nice addition to the published literature.

Reviewer #3: In this manuscript, the authors present a computational model for exosome-mediated chemotaxis. Analysis of the model reveals that it produces a non-monotonic dependence of the migration velocity on the exosome cargo size. The modeling approach and analysis seem technically sound. However, my concern with the paper is that I don’t see any evidence in the references cited by the authors for cells detecting chemical gradients through the uptake of diffusing exosomes. And it is not clear how this would work. Chemical gradients are detected by receptors located on the cell surface. If the contents of exosomes are taken up by the cell, how is the cell able to use this information to detect a gradient? One potentially interesting role for exosomes in gradient sensing has been suggested by Arya et al. (Nat. Cell Bio., 2025, 27:931). In their model, DNA secreted from neutrophils ensnares LTB4-containing exosomes. This creates a spatial gradient of LTB4 that can be detected by migrating neutrophils and macrophages. I think this would be an interesting scenario to model to investigate if it provides advantages over the direct release of LTB4.

**Have the authors made all data and (if applicable) computational code underlying the findings in their manuscript fully available?**

Reviewer #1: Yes

Reviewer #2: Yes

Reviewer #3: Yes

PLOS authors have the option to publish the peer review history of their article (what does this mean?). If published, this will include your full peer review and any attached files.

Reviewer #1: No

Reviewer #2: No

Reviewer #3: No

**Figure resubmission:**
---

## [Decision Letter · Decision Letter 1]

6 Jan 2026

Dear Mugler,

We are pleased to inform you that your manuscript 'Exosome-mediated chemotaxis optimizes leader-follower cell migration' has been provisionally accepted for publication in PLOS Computational Biology.

Best regards,

Sunil Laxman, PhD

Academic Editor

PLOS Computational Biology

Marc Birtwistle

Section Editor

PLOS Computational Biology

Reviewer's Responses to Questions

**Comments to the Authors:**

Reviewer #1: I think the authors have addressed the comments raised by the referees, and the manuscript is now suitable for publication in PLOS Computational Biology.

Reviewer #2: I thank the authors for their scholarly and thorough response. They have completely addressed the points raised, and in my opinion this manuscript can now be published as-is. I congratulate the authors on this fine piece of work.

**Have the authors made all data and (if applicable) computational code underlying the findings in their manuscript fully available?**

Reviewer #1: Yes

Reviewer #2: Yes

PLOS authors have the option to publish the peer review history of their article (what does this mean?). If published, this will include your full peer review and any attached files.

Reviewer #1: No

Reviewer #2: No

---

## [Editor Report · Acceptance letter]

PCOMPBIOL-D-25-02085R1

Exosome-mediated chemotaxis optimizes leader-follower cell migration

Dear Dr Mugler,

I am pleased to inform you that your manuscript has been formally accepted for publication in PLOS Computational Biology. Your manuscript is now with our production department and you will be notified of the publication date in due course.

With kind regards,

Anita Estes
